# Climate-Responsive Green-Space Design Inspired by Traditional Gardens: Microclimate and Human Thermal Comfort of Japanese Gardens

Lihua Cui [1,*], Christoph D. D. Rupprecht [2] and Shozo Shibata [3]

1   Graduate School of Agriculture, Kyoto University, Kyoto 606-8502, Japan
2   FEAST Project, Research Institute for Humanity and Nature, Kyoto 603-8047, Japan; crupprecht@chikyu.ac.jp
3   Graduate School of Global Environmental Studies, Kyoto University, Kyoto 606-8501, Japan; shibata.shozo.6n@kyoto-u.ac.jp
*   Correspondence: lihua.cui.36x@st.kyoto-u.ac.jp

**Abstract:** Urban green spaces can provide relaxation, exercise, social interaction, and many other benefits for their communities, towns, and cities. However, green spaces in hot and humid regions risk being underutilized by residents unless thermal environments are designed to be sufficiently comfortable. Understanding what conditions are needed for comfortable outdoor spaces, particularly how people feel in regard to their thermal environment, is vital in designing spaces for public use. Traditional gardens are excellent examples of successful microclimate design from which we can learn, as they are developed over the generations through observation and modification. This study analyzed how Japanese gardens affect people's thermal stress on extremely hot summer days. Meteorological data was collected in three Japanese gardens, and human thermal comfort was evaluated through physiological equivalent temperature (PET). Statistical analysis examined the relationship between spatial configurations of the gardens and thermal comfort. Our study revealed that Japanese gardens can efficiently ameliorate thermal stress. Spatial analysis showed that garden elements affect thermal comfort variously depending on time of the day and spatial distribution.

**Keywords:** Japanese garden; microclimate; thermal comfort; climate responsive strategy; PET; urban green-space design; human well-being; comfortable urban environment

## 1. Introduction

### 1.1. Urban Green Space and Human Well-Being

Interacting with nature and being exposed to greenery is an innate desire of human beings [1]. Strong evidence shows that people exposed to more greenery are healthier, both physically and psychologically [2]. However, in our globally urbanizing world, green space for outdoor activities is declining and with it time people spend in it [3–6]. Apart from the health concerns, an "extinction of experience" has been identified to occur worldwide, and studies demonstrated that people who spend less time in nature have less knowledge of or interest in nature [7–9]. This is particularly obvious among children, who today spend less time outdoors than previous generations. Soga and colleagues [10] warn that more people will turn biophobic if the current trend continues, which is likely to affect future nature and environmental conservation [11].

Green-space accessibility is one important factor influencing whether people visit and use them. Numerous studies have proven that people who have better access to green spaces engage more frequently in outdoor activities [12–14]. Providing sufficient green spaces, both in terms of number and area, has become a crucial criteria to assess if a city is healthy and sustainable. For example, some cities and countries set a target of minimum green spaces per person or set a minimum distance from residents' homes to urban parks. However, sufficient quantities of green spaces alone are not enough to

ensure high frequency of outdoor activities or improved human well-being [15,16]. On the contrary, urban green spaces that are underutilized are more likely to become crime scenes or to be perceived as economic and social burdens [17–19]. Therefore, scholars emphasize that to improve people's spontaneous outdoor activities in urban green spaces, both quantity and quality of green spaces should be kept in mind when urban planners and green-space designers create urban green spaces [20,21].

*1.2. Climate Responsive Green Space Strategies*

Thermal comfort is one of the qualities of urban green spaces that can significantly affect people's willingness to spend time there. Studies investigated people's green-space use revealed that thermal conditions strongly influence the number of participants and length of use [22–25]. For instance, in urban green spaces in Tokyo, people stayed for more than 30 min only if short radiation was lower than 200 W/m$^2$, and people stayed longer when the wind speed was around 0.5 m/s and the standard effective temperature (SET) was around 30 °C [25]. These studies indicate a green space with comfortable thermal conditions is a more livable and beneficial green space. However, in regions that have an uncomfortable prevailing climate, such as a hot and humid climate, creating pleasant thermal environments outdoors is not an easy task, or as simple as just planting some trees.

The fact that urban green spaces have cooling effects has been commonly accepted by urban planners and researchers, with ample evidence from many parts of the world. Spronken-Smith and Oke [26] indicated that parks in Vancouver are typically 1–2 °C cooler than their surroundings. In Utrecht, the difference in maximum air temperature between the city center and 13 parks varied from 0.3 °C to 1.5 °C in summertime [27]. Yet some researchers suggested that green spaces do not necessarily work to improve people's thermal comfort. A study conducted in Tokyo suggested that urban parks are not always beneficial in terms of human thermal comfort due to ineffective placement and design [28]. Another study conducted in Taipei, Chang et al. [29] found one-fifth of 61 urban parks observed were warmer than their urban surroundings on summer days. In a comprehensive review on thermal comfort of green spaces [30], Shooshtarian et al. argued that green spaces generally improve thermal conditions, and accordingly the thermal experience of people, but the effects vary by sites and occasions. Moreover, people's preferences to thermal conditions appear varied across different regions. Lin et al. [23] and Thorsson et al. [24] respectively reported 70% and 80% of people in urban parks situated under shade in southern Taiwan and Tokyo in summer days. However, in Sweden more people enjoyed the sun in summer, while only 14% of people were under shade in urban parks in Göteborg [31]. Previous studies indicated there are no universally applicable comfortable green-space designs. Therefore, it is important to determine climate-responsive green-space strategies on a site-by-site basis, and the strategies should be examined carefully prior to advising decision-makers.

*1.3. Microclimate of Traditional Gardens*

Learning from successful cases likely represents an efficient way to gain insight into climate-responsive green-space strategies [32], particularly because testing cooling effects of various green-space forms under different environments can be time-consuming and costly. Several good cases in one region provide a valuable opportunity to identify effective strategies by analyzing their designs. Traditional gardens such as Persian gardens [33,34] and Chinese gardens [35,36] have been seen as good examples of comfortable green spaces and microclimates. Nevertheless, in general, the number of studies about the traditional garden microclimate is severely limited compared to studies of microclimates in urban parks. Moreover, to our knowledge there is no study about the Japanese garden microclimate. In this exploratory research, we sought to improve understanding of climate-responsive strategies by studying traditional gardens in the historical city Kyoto, Japan.

The former capital city of Japan, Kyoto is the hub of Japanese culture and traditions, with a history reaching back more than 10 centuries before 1869, when the imperial court

relocated to Tokyo. The first form of gardens in Japan emerged in the 7th century when a strong influence of arts from China reached Japan. Over a long period of time, gardening skills were greatly developed, and by the Muromachi period (14–16th century), garden style in Japan had become mature, with its own philosophy and aesthetics. The Edo period (17–19th century) saw the greatest number of Japanese gardens being constructed, and many historical gardens that remain today were mostly built in this period. The more uncomfortable the thermal condition of the place, the more knowledge and wisdom about climate-responsive adaptations was accumulated [37]. The hot and humid climatic conditions of Kyoto required developing a certain gardening strategies able to adapt to the local climate and improve living conditions.

### 1.4. Research Questions and Objectives

This study seeks to explore the hypothesis that the spatial designs of Japanese gardens not only reflect a balanced proportion and function, but were also developed to be thermally comfortable. For this purpose, an experimental study was designed to identify local and climate-responsive green-space-design strategies from the historical gardens of Kyoto, driven by the following research questions:

1. Are historical/traditional gardens in Kyoto thermally comfortable?
2. How do the gardens ameliorate human thermal comfort? Does a certain configuration of the gardens modify the microclimate? Do certain garden elements play a critical role in modifying the microclimate?
3. How might heat-mitigation strategies of Japanese gardens be applied to urban green spaces?

## 2. Materials and Methods

For this study, three traditional Japanese gardens were selected to analyze how the traditional gardens modify human thermal comfort in hot summer days in Kyoto. Meteorological data were collected in the gardens, and spatial configuration of the gardens were analyzed. Human thermal comfort was evaluated using the physiological equivalent temperature (PET) index. Lastly, the relationship between thermal comfort and garden configurations was analyzed statistically.

### 2.1. Study Site

#### 2.1.1. Overview of Kyoto

The study site of Kyoto ($35°0'42''$ N, $135°46'6''$ E) is located in a basin surrounded on three sides by mountains, with the south side opening to the Osaka plain. The climate of Kyoto is humid subtropical (Cfa, Köppen climate) with an annual precipitation of 1491.3 mm, a mean annual temperature of 15.9 °C, and the daily mean highest temperature in August of 33.3 °C (observed values from 1981 to 2010, released in 2011) [38]. Summer starts after the approximately one-month-long rainy season lasting until late July, and is characterized by extreme humidity and high temperature until late August. Kyoto is not the hottest city in Japan, but ranks at the top for number of extremely hot days (defined as days with temperature exceeding 35 °C) among the cities in Japan [38].

As a historical city, the city features narrow streets and dense construction with high population density. Today's heavy traffic and pervasive air conditioning exacerbate the urban-heat-island effects and negatively affect thermal perception. Global warming and climate-change-related extreme weather events are bound to further impact the living experience and human well-being. Evidence that climate change affects human well-being in Kyoto has become manifest in the form of continuous years of extreme weather events in summer. Over the last few years, both the number of extreme hot days and the number of people who suffered heatstroke increased in Kyoto [38,39]. The Japan Meteorological Agency [38] predicts this trend will continue, compounding the urgency to enact efficient climate-responsive strategies to protect people from summer heat and provide safe and high-quality green spaces in urban areas.

### 2.1.2. Studied Gardens in Kyoto

The studied gardens were determined according to the following conditions: (1) gardens near to each other to minimize the difference in micrometeorological conditions across the study sites, (2) gardens of similar scale, and (3) gardens constructed in the same historical time period. After careful selection, the study area was narrowed down to the Higashiyama area, located in the eastern hillside of Kyoto. The area is emblematic of the tradition of Kyoto, where people with wealth and power enthusiastically built gardens in their grounds as a space for ritual events, guest reception, meditation, and self-relaxation. Many historical gardens in Higashiyama area are still well preserved today.

Shoren-in garden (Garden S), Chion-in garden (Garden C), and Joju-in garden (Garden J) were chosen as study sites (Figure 1). All of the three gardens are designated cultural assets and are under protection (Table 1). Shoren-in is open for public visits, while the other two are not regularly opened, but occasionally public events and ritual events are organized in all three gardens. Preservation-related restrictions implied certain restrictions for data collection, which we discuss in more detail in Section 4.4.

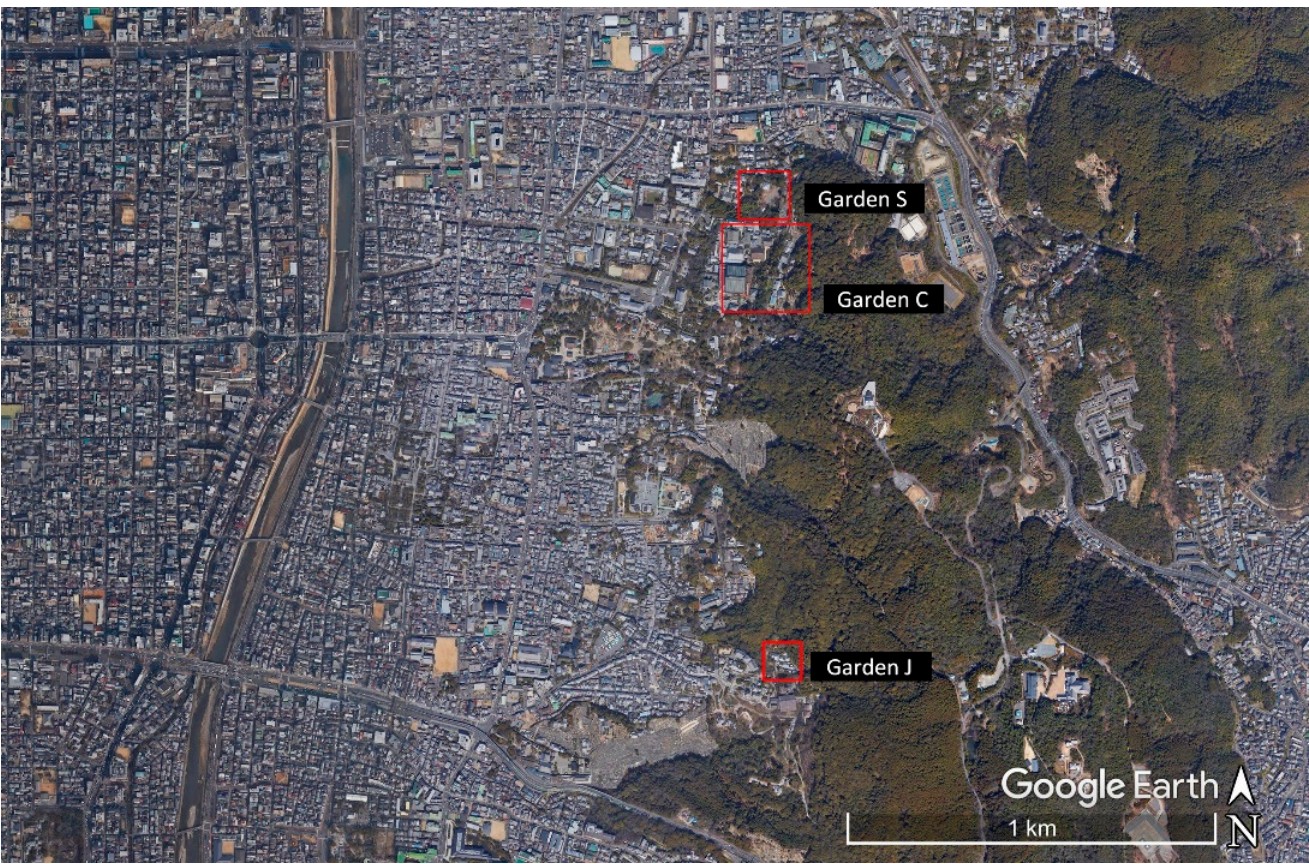

**Figure 1.** Location of the gardens in Higashiyama Ward in Kyoto, Japan.

### 2.2. Meteorological Data Collection and Thermal-Comfort Measurement

Data Collection at the Study Sites

In each garden, two thermometers were set in two areas (Figure 2). One thermometer was used in the garden while the other thermometer was used in open areas outside of the garden. The three gardens are sedentary gardens, meaning people do not walk into the gardens but sit at the resting areas of opened halls surrounding the gardens. To understand how people feel thermally when they watch the garden, investigation points were set at the resting areas of the halls that surround the gardens. No investigation point was set in the gardens. To understand the differences in meteorological conditions and thermal

comfort between the gardens and surrounding urban open space, study points were set at open areas near the gardens. In the open areas the study points were set at points less likely to be influenced by vehicles and less likely to be overshadowed by nearby trees or buildings (Figure 3).

**Table 1.** Profile of the study sites.

|  | Garden S | Garden C | Garden J |
|---|---|---|---|
| Construction of the garden * |  | 17th century |  |
| Garden style | Pond garden Sedentary appreciation | Pond garden Sedentary appreciation | Pond garden Sedentary appreciation |
| Special status | Has a close relationship with the emperor family; once was the residence of Empress Go-Sakuramachi | Municipal place of scenic beauty | National place of scenic beauty |
| Area | 0.21 ha | 0.40 ha | 0.13 ha |
| Canopy coverage | 52.4% | 70.7% | 38.5% |
| Pond area ratio | 8.1% | 21.8% | 12.3% |
| Dominant arbor tree species | *Acer palmatum, Ternstroemia japonica* | *Acer palmatum, Castanopsis cuspidata* | *Pinus densiflora* |
| Dominant shrub species | *Rhododendron indicum* | *Rhododendron indicum* | *Rhododendron indicum* |
| Dominant green ground materials | Lawns | Moss | Lawns |

* Exact years of construction of the three gardens are unclear, but the gardens are commonly accepted as Edo-period gardens according to their landscaping features. General structures of the gardens were estimated to be completed in the 17th century according to historical literature and paintings [40].

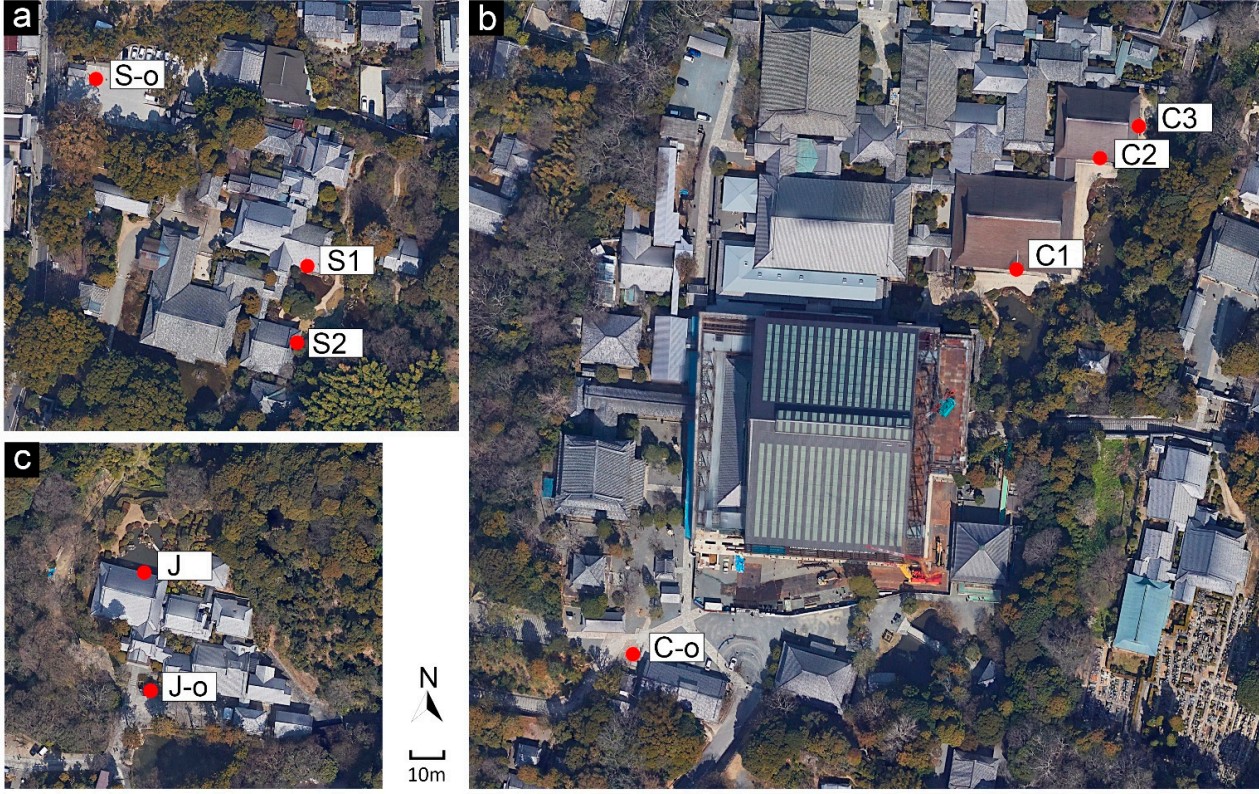

**Figure 2.** Study points in the three gardens and the open areas near them: (**a**) Garden S, S1, and S2 represent study points in resting areas, and S-o represents the open area (likewise for Garden C and Garden J), (**b**) Garden C, and (**c**) Garden J.

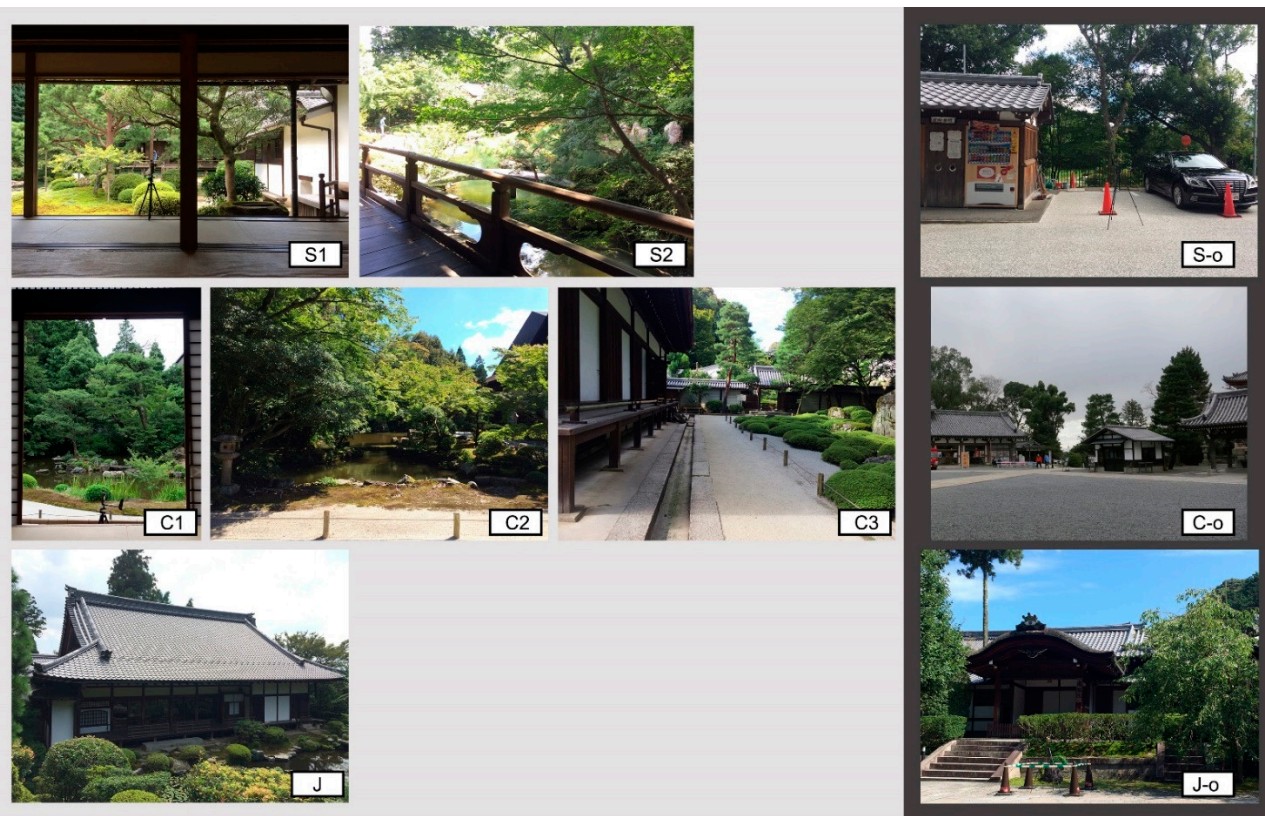

**Figure 3.** Study sites in the three gardens (light grey background) and the open areas near them (dark grey background).

Air temperature (*Ta*), globe temperature (*Tg*), relative humidity (*RH*), and wind velocity (*v*) were collected at the study points with thermometers (Kestrel 5400, USA). The height of the thermometers was set to 0.5 m in the resting areas and 1.5 m in the open areas, which is about the height of people sitting on a floor and standing (Figure 3). The reason for selecting the Kestrel 5400 as the instrument was its ability to collect all the parameters required for calculating the thermal perception index PET with one small device. It is noteworthy that the Kestrel 5400 is not radiation-shielded or as sensitive as other more sophisticated weather stations, but we deemed it the best option for carrying out experiments in these gardens, since installing full weather stations in the gardens was not permitted.

The study was conducted for 3 days in each garden for a total data collection of 9 days in August 2017. The instrument in the gardens was moved from point to point in Garden S and Garden C, as there was more than one resting area. The survey was designed to collect a set of data in every 30 min; accordingly, in Garden S the thermometer was moved every 15 min between 2 study points and in Garden C every 10 min among 3 study points. All surveys were conducted from 9:00 to 16:00 (see Supplementary File S2). The dates were determined according to the weather conditions (clear sky or party clouded) and the availability of the gardens (Table 2).

**Table 2.** Survey schedule and mean of weather conditions on the survey dates in August 2017 [38].

| Date | 2nd | 3rd | 4th | 6th | 9th | 12th | 13th | 14th | 17th | Ave. |
|---|---|---|---|---|---|---|---|---|---|---|
| Garden | J | C | S | J | S | C | J | S | C | |
| Air temperature (°C) | 27.3 | 29.9 | 29.4 | 29.1 | 28.1 | 28.2 | 27.3 | 29.0 | 29.9 | 29.8 |
| Highest Air temperature (°C) | 31.6 | 35.7 | 33.5 | 34.5 | 32.7 | 32.1 | 32.1 | 35.1 | 36.7 | 33.8 |
| Relative Humidity (%) | 62 | 63 | 68 | 70 | 66 | 64 | 64 | 63 | 71 | 65.7 |

In Garden S, there are two halls: one is in the north of the garden and called Kachoden (S1), the other one is called Kogo-sho (S2) and is located in the south of the garden. S1 is relatively open and bright, as there is no tree taller than the roof near the hall. In comparison, S2 is heavily shaded by tree canopy on its eastern side, and only its northern side is open to the garden. A pond is located in the eastern part of the garden, and a stream flows down from the south and into the pond and to the north. Behind the pond is a hill creating a green background for the garden. Chion-in also has two halls for watching the garden located in the west, a pond is located on the south-eastern part of the garden, and next to the pond is a hill that is completely forested. The bigger hall Ohojo (C1) faces the pond on its south, and is an open and bright hall with only a few trees near. The smaller hall is called Kohojo (C2) and is located on the northeastern side of C1, facing a pond on its south as well. On the east of the Kohojo (C3) is a small garden paved with moss and Azalea, and thickly covered by trees. Joju-in (J) is a tranquil garden surrounded by greenery on three sides except in the south, where the only hall is standing facing the pond on its north. The hall is relatively dark as it is facing north, with no big trees near the hall (Figure 3).

### 2.3. Human Thermal Comfort Calculation—PET

Physiological equivalent temperature (PET) was used to evaluate the thermal comfort of the gardens. PET was developed by Hoppe [41] in the 1990s according to the Munich Personal Energy Balance Model, and today is the most widely used index to assess outdoor thermal perception. The thermal condition in PET is defined as the air temperature at which, in a typical indoor setting ($Tmrt = Ta$; $VP = 12$ h, $v = 0.1$ m/s), the heat budget of the human body is in equilibrium with the same core and skin temperatures as those under complex outdoor conditions. RayMan model 1.2 was used to calculate PET [42]. This requires input of the mean radiant temperature ($Tmrt$), air temperature ($Ta$), relative humidity ($RH$), and wind speed ($v$) as meteorological parameters, as well as personal parameters and clothing-insulation data. The metabolic rate was set to a sitting-quietly level (58 W). The clothing insulation was set to typical summer clothes (0.5 clo), and the personal data was set to a typical Japanese man at 35 years old (1.7 m tall and 70 kg). $Tmrt$ was calculated using the formula below (based on ISO 7726 standard):

$$Tmrt = [(273 + Tg)^4 + 1.10 \times 10^8 v^{0.6}(Tg - Ta)/\varepsilon D^{0.4}]^{0.25} - 273 \tag{1}$$

where $Tg$ is globe temperature, $Ta$ is air temperature, $v$ is wind velocity, $\varepsilon$ is the emissivity of the globe (0.95), and D is globe diameter (0.025 m) of globe thermometer.

Comfortable range in PET was assessed based on the questionnaires to identify which thermal condition was acceptable for 80% to 90% of people and could thus be understood as comfortable. In Central Western Europe, the PET comfortable range is 18–23 °C [43]. While Lin and Matzarakis [44] obtained a range of 26–30 °C PET at intervals of 4 °C PET as a more reasonable comfort range in southern Taiwan, in a hot and humid climatic context. In this study, the thermal-comfort range assessed by Lin and Matzarakis was adopted because the climate of Kyoto, being humid and hot in summer, is closer to southern Taiwan than Central Western Europe (Oceanic climate).

### 2.4. Garden-Configuration Analysis

The area of typical garden elements in Japanese gardens, including tree canopy, green ground, pavement, open water, and roof, were measured by AutoCAD LT 2019 within the range of 5 m, 10 m, and 20 m radius from the study points (Figure 4, Table 3). As shading is suggested to be critical to human thermal comfort [45], the area covered by either tree canopy or roof was additionally measured and separated in four directions: north, east, south, and west. The boundary of the garden elements in the three measurement radii was distinguished based on Google Earth Pro (aerial photography taken in May 2017), site plans of the gardens, and onsite confirmation (see Supplementary File S1).

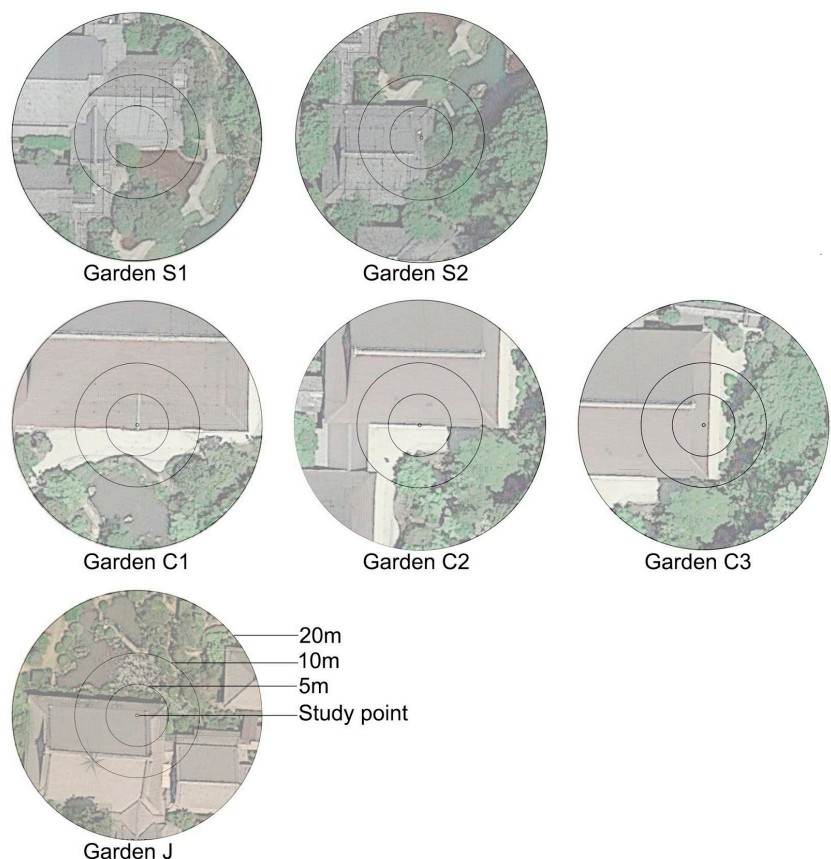

**Figure 4.** Spatial configurations in the three measurement radii from the study points.

**Table 3.** Definition of garden elements and examples of garden-elements boundaries within a 20 m radius from the study points, taking the study point S1 as an example.

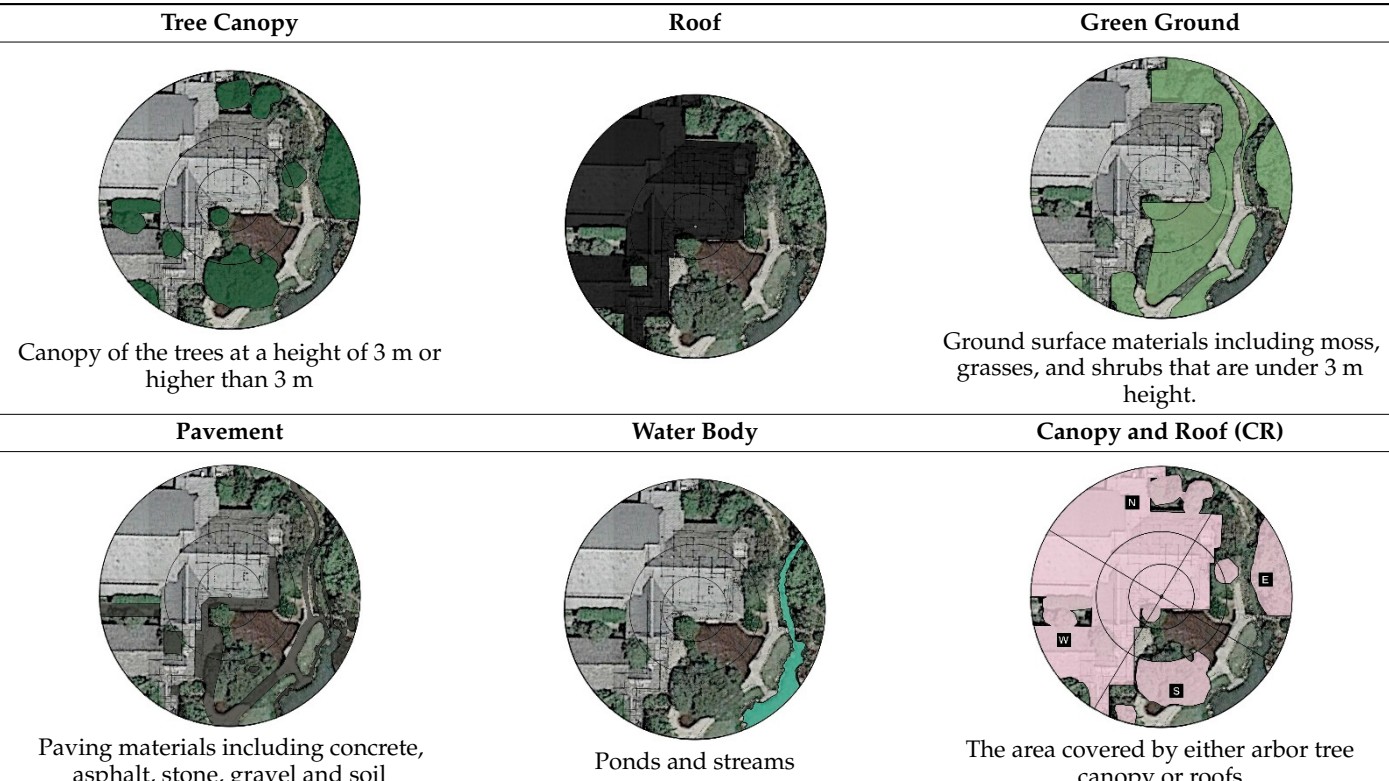

| Tree Canopy | Roof | Green Ground |
|---|---|---|
| Canopy of the trees at a height of 3 m or higher than 3 m | | Ground surface materials including moss, grasses, and shrubs that are under 3 m height. |
| **Pavement** | **Water Body** | **Canopy and Roof (CR)** |
| Paving materials including concrete, asphalt, stone, gravel and soil | Ponds and streams | The area covered by either arbor tree canopy or roofs |

### 2.5. Statistical Analysis

A Kruskal–Wallis one-way analysis of variance with Dwass–Steel–Critchlow–Fligner pairwise comparisons was conducted to test for differences in meteorological conditions and thermal comfort in the gardens and the corresponding open areas. The Pearson correlation coefficient was calculated to determine the relationship between PET and the ratio of the garden elements within different ranges of radius and in different ranges of time. All statistical analysis was conducted with R, version 4.0.2 [46]. The threshold for significance was set at $p = 0.05$.

## 3. Results

### 3.1. Microclimate of the Study Sites

Table 4 shows mean differences in meteorological conditions between the studied open areas and the corresponding gardens. Compared with the open areas, the gardens had on average 1.35 °C lower *Ta*, 8 °C lower *Tg*, approximately 5% higher *RH*, and less air movement.

**Table 4.** Mean differences in meteorological parameters between open areas and corresponding gardens.

|  | S1 | S2 | C1 | C2 | C3 | J | Average |
|---|---|---|---|---|---|---|---|
| Air temperature (°C ) | 2.22 | 2.78 | 0.03 | 0.17 | 1.02 | 1.91 | 1.35 |
| Globe temperature (°C ) | 8.43 | 9.38 | 5.40 | 6.59 | 8.30 | 9.62 | 7.95 |
| Relative humidity (%) | −9.69 | −9.82 | −0.92 | −1.01 | −3.79 | −4.92 | −5.03 |
| Wind speed (m/s) | 0.48 | 0.20 | 0.69 | 0.85 | 0.78 | 0.39 | 0.57 |

Among the study points in the open areas, the highest mean *Ta* was 32.5 °C in S-o, and the lowest mean *Ta* was 31.2 °C in C-o. In the three gardens, the highest mean *Ta* was 31.1 °C in C1 with nearly no difference from the C-o. The lowest mean *Ta* was 29.7 °C in S2. The mean *Ta* of the points in Garden S and Garden J were all significantly lower than the corresponding open areas, but no statistical difference was found between the mean *Ta* of C1, C2, and C-o, and only that of C3 was significantly lower than that of C-o (Figure 5a).

A similar pattern was found in mean *RH* (Figure 5c). Mean *RH* in S-o and J-o were significantly lower than any of the study points in the corresponding gardens, but in Garden C, there was only a significant difference between C1, C3, and C-o, but no difference between C2 and C-o. S-o had the lowest mean *RH* at 56%, and J-o had the highest mean *RH* at 62.8%. The difference in *Tg* between the open areas and the gardens was most pronounced (Figure 5b). The highest mean *Tg* was 40.4 °C in J-o, and the lowest mean *Tg* was 39.4 °C in S-o. The highest mean *Tg* in the gardens was 34.2 °C in C1, where the environment is brightest among the studied halls. S2 ranked as the lowest mean *Tg*, at 30 °C.

As Figure 5d shows, mean *v* in the three open areas was much higher than those in the gardens, and air movement was particularly frequent in C-o with the mean *v* 1.05 m/s. Mean *v* in the gardens were all lower than 0.4 m/s. Mean *v* in S2, C1, and C3 were relatively higher than the other three study points, but the median *v* of the study points in the gardens were all 0 m/s except for C3, with a median *v* of 0.2 m/s.

When comparing the meteorological conditions among the study points in the three gardens, significant differences were also found. In Garden S, S1 was significantly lower than S2 in *Ta* and *Tg*, but higher in *v*. In Garden C, the microclimates of C1 and C2 were identical, as no difference was found in the four atmospheric parameters between the two resting areas. In comparison, the differences between C3 and the other two resting areas were relatively more pronounced, particularly in *Tg*, as *Tg* in C3 was significantly lower than the other two study points.

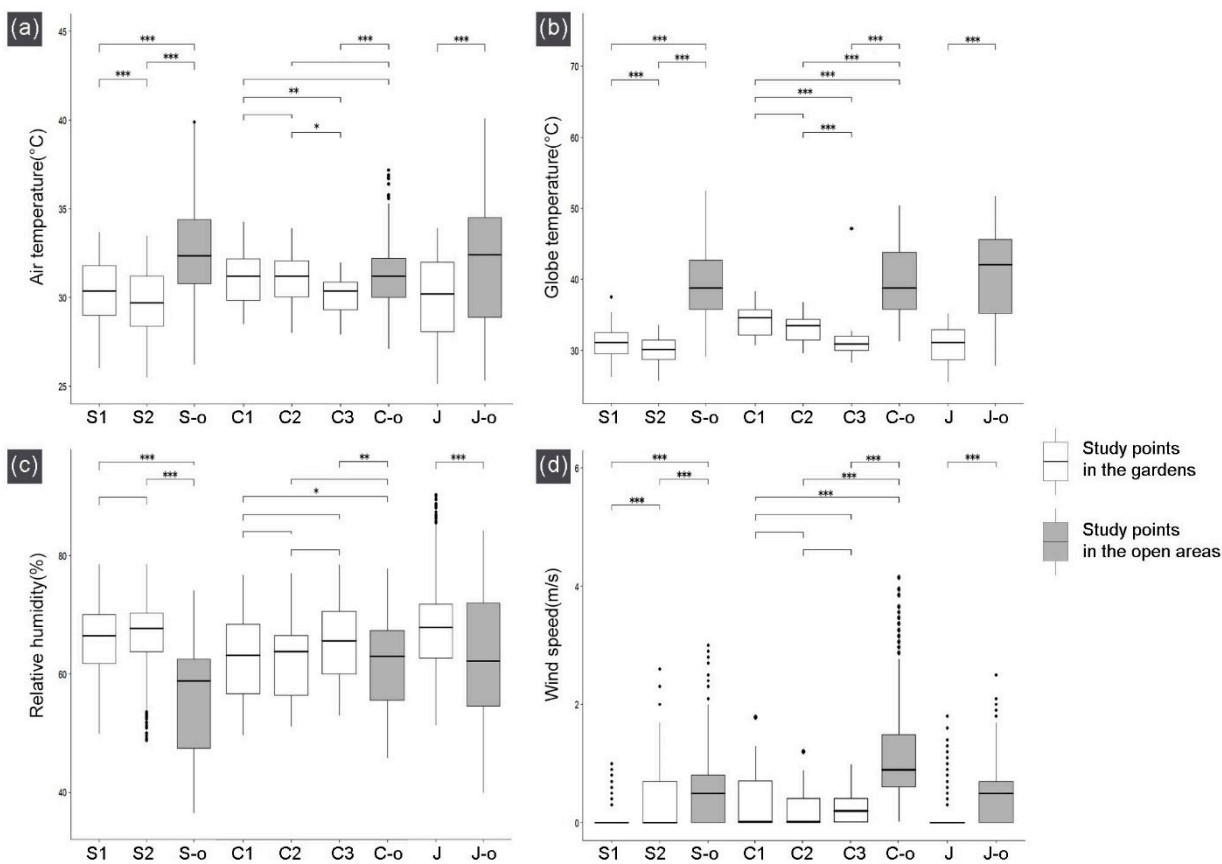

**Figure 5.** Mean values of meteorological parameters in gardens and surrounding open areas during the survey, observed from 9:00 to 16:00. Pairwise comparisons were carried out among the study points in the gardens and the corresponding open areas; * $p < 0.05$, ** $p < 0.01$, *** $p < 0.001$.

### 3.2. Human Thermal Comfort of the Study Sites

The PET value of the open areas highlights the uncomfortable thermal condition during summer in Kyoto (Figure 6). Most of the PET values of the three open areas were "hot" or "very hot" during the survey days. Among the study points in the open areas, C-o recorded the highest mean PET of 49.2 °C, followed by J-o at 45.8 °C, while the lowest mean PET was 44.5 °C in S-o. In contrast, thermal conditions in the gardens were mostly "slightly warm" (PET range: 30–34 °C). The mean PET values of all points except C1 were in the "slightly warm" range; the lowest mean PET was in S2 with 30.5 °C, while the highest mean PET was in C1 with 35.7 °C. When comparing the mean PET among the study points in the three gardens, C1 and C2 were the significantly highest ones, and similar to the atmospheric parameters, there was no significant difference in PET between C1 and C2. S1, C3, and J had similar thermal-comfort conditions. The resting areas of them were mostly "Slightly warm", but the thermal condition could turn "Warm" sometimes. S2 had the lowest PET, and its mean PET was significantly lower than that of any other study points. During the survey, the resting area of S2 was assessed as "Comfortable" for half of the time and "Slightly warm" for the rest of the time. Interestingly, study points in the same garden were only a short distance apart, yet the thermal conditions could be significantly different; namely S1 and S2, and C1, C2, and C3.

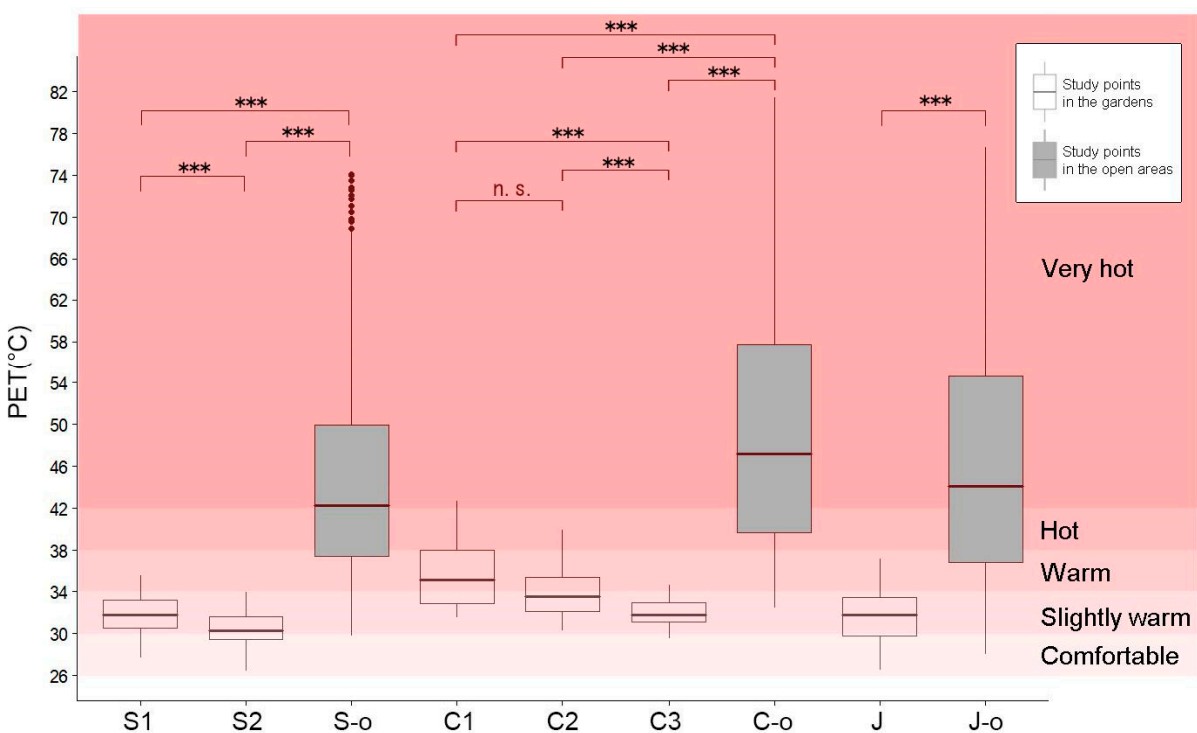

**Figure 6.** Mean PET values in the gardens and the surrounding open areas during the survey, observed from 9:00 to 16:00. Pairwise comparisons were carried out among the study points in the gardens and the corresponding open areas; *** $p < 0.001$.

### 3.3. Garden Spatial Configurations

Trees are one of the important garden elements that shape the gardens' spatial configuration. Spatial configurations around the six resting areas that we investigated could be classified into two patterns according to the tree plantation (Figure 7a). One is the presence of dense forest near the areas, resulting in the areas being well shaded by the tree canopy, such as S2 and C3. The other pattern is the presence of only a few trees near the resting areas, with the trees being planted for creating a scenic view rather than for creating the shade. S1, C1, C2, and J are areas that have fewer trees around them. Especially in C1, C2, and J, there are no trees around the resting areas within a 5 m radius.

Both tree canopy and roof are elements effective at blocking shortwave radiation. In S2 and C3, with their large canopy (29.7% and 20.9% on average, respectively) and roofs on the south and west sides of the resting areas, the areas enjoy shade for most of the time during the day. The study point in Garden J does not have a large canopy around (5.4% on average), but the resting area is facing north, hence little solar radiation directly affects people's thermal comfort. S1, C1, and C2 are open to the south and with limited trees near these areas (canopy ratio, 12.0%, 6.5%, and 13.1% on average, respectively), the environment of these three areas is very bright during the day.

Figure 7b displays the patterns of the ground-surface materials' distribution around the resting areas. The ratio of green ground was relatively small within a 5 m radius, but it increased with range. Within a 20 m radius, the ratio of green ground was 30.1% on average. In contrast, the ratio of pavement was largest in the 5 m range (27.6% on average), but smallest in the 20 m radius, at 16.8% on average. Regarding the water-body distribution, S2 and J were the only two resting areas where the areas were very close to the ponds (24.5% and 10.0%, respectively, within a 5 m radius). Except for S2 and J, other resting areas were some distance from the pond, and the ratio of the water (7.9% on average in 20 m radius) was generally lower than the ratio of green ground and pavement.

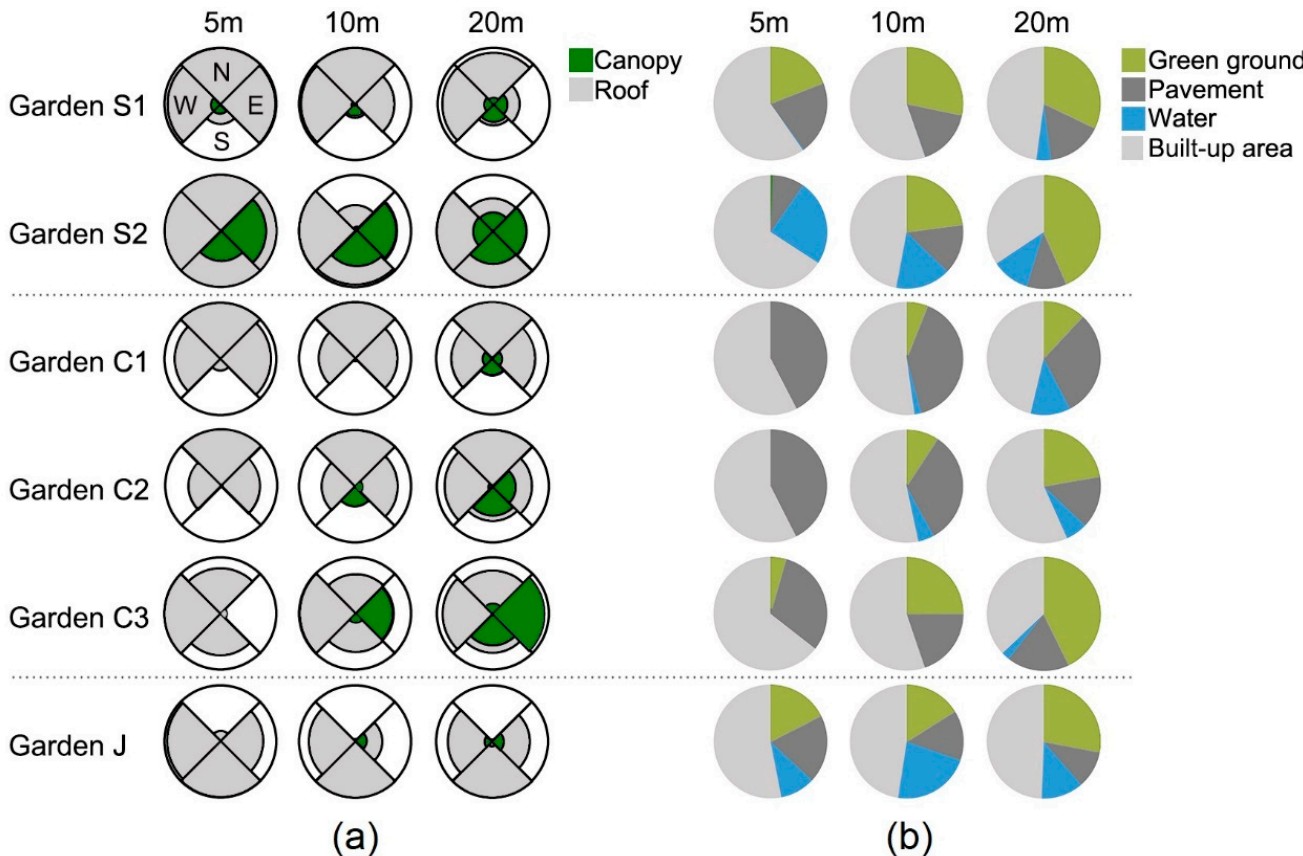

**Figure 7.** (**a**) Ratio of tree canopy and roofs across the six study points within 5 m, 10 m, and 20 m radius, calculated and presented separately for four directions. (**b**) Ratio of ground-cover elements around the study points within 5 m, 10 m, and 20 m radii.

### 3.4. Relationship Between Garden Configurations and Human Thermal Comfort

The five garden elements (canopy, roof, green ground, pavement, and water) all affected thermal comfort of resting areas significantly, although the effects found in this study were generally limited (Table 5). The degree of effects varied across different distance ranges and different times of day. Table 5 shows the correlation between the ratio of garden elements and PET in three different radii. Canopy significantly affected thermal comfort but the effect was small in all three radii ($r < |0.2|$, $p < 0.001$). Water mildly modified thermal comfort, but only within the 5 m range ($r = -0.241$, $p < 0.001$), while the effect became very weak in larger ranges. Pavement was the most significant influencer and a negative influence on thermal comfort ($r = 0.334$, $p < 0.001$, within 5 m radius). Roofs had mild cooling effects only within the 5 m range ($r = -0.208$, $p < 0.001$). However, a larger area of roofs in a large range negatively affected thermal comfort. The relationship between green ground and PET only became significant within 20 m radius ($r = -0.25$, $p < 0.001$). Canopy and roof (CR) improved the surrounding thermal environment mildly within 5 m range ($r = -0.207$, $p < 0.001$). When examining the connection between CR and thermal comfort in different directions, only CR on the west within the 5 m and 10 m ranges reduced thermal stress to a significant and mild degree ($r = -0.25$ and $r = -0.228$, respectively, $p < 0.001$). CR on the south of the halls also significantly reduced thermal stress, but to limited effects. CR on the north and east side of resting areas generally had a negative though weak effect on thermal comfort.

**Table 5.** Pearson's correlation between the ratio of garden elements and PET within 5 m, 10 m, and 20 m radii.

| Garden Element | 5 m | 10 m | 20 m |
|---|---|---|---|
| Canopy | −0.188 *** | −0.174 *** | −0.103 *** |
| Green ground | −0.016 | −0.147 *** | −0.25 *** |
| Pavement | 0.334 *** | 0.303 *** | 0.255 *** |
| Water | −0.241 *** | −0.122 *** | −0.068 ** |
| Roof | −0.208 *** | 0.113 *** | 0.203 *** |
| CR | −0.207 *** | −0.087 *** | −0.007 |
| CR.N | 0.023 | 0.11 *** | 0.092 *** |
| CR.E | −0.104 *** | 0.027 | 0.105 *** |
| CR.S | −0.197 *** | −0.176 *** | −0.164 *** |
| CR.W | −0.25 *** | −0.228 *** | −0.162 *** |

$p < 0.01$ **, $p < 0.001$ ***; light grey highlighted figures indicate a significantly mild correlation ($|0.2| \leq r < |0.4|$).

During the day, the microclimate in the gardens changed with the time as well as the thermal conditions. Therefore, we additionally assessed the cooling effects of the garden elements in different time ranges by testing the correlation between PET in the morning (9:00–11:00), noon (11:00–14:00) and afternoon (14:00–16:00) and garden elements ratio in the 5 m, 10 m, and 20 m radii.

3.4.1. Relationship between Garden Configurations and Human Thermal Comfort in the Morning

The three gardens are located at the foot of mountains on the east side of the city. Most of the solar radiation in the morning is thus blocked by the hill, resulting in a relatively cooler thermal conditions. Table 6 shows the cooling effects of canopy and CR were not significant; only the CR on the east and close to resting areas had mild effects on PET (r = −0.206, $p < 0.001$). Water significantly and mildly affected thermal comfort in the 5 m range (r = −0.248, $p < 0.001$) but the effect was nonsignificant in the larger ranges. Pavement negatively affected thermal perception in all three ranges (r = 0.303, $p < 0.001$, in 5 m radius).

**Table 6.** Pearson's correlation between ratio of garden elements and PET in the morning.

| Garden Element | 5 m | 10 m | 20 m |
|---|---|---|---|
| Canopy | −0.144 *** | −0.095 ** | −0.004 |
| Green ground | −0.051 | −0.031 | −0.109 * |
| Pavement | 0.303 *** | 0.254 *** | 0.272 *** |
| Water | −0.248 *** | −0.19 *** | −0.169 *** |
| Roof | −0.194 *** | 0.131 ** | 0.113 ** |
| CR | −0.168 *** | 0.011 | 0.078 |
| CR.N | 0.075 | 0.15 *** | 0.141 *** |
| CR.E | −0.206 *** | 0.075 | 0.15 *** |
| CR.S | −0.188 *** | −0.184 *** | −0.187 *** |
| CR.W | −0.14 *** | −0.087 * | −0.079 |

$p < 0.05$ *, $p < 0.01$ **, $p < 0.001$ ***; light grey highlighted figures indicate significantly mild correlation ($|0.2| \leq r < |0.4|$).

3.4.2. Relationship between Garden Configurations and Human Thermal Comfort at Noon

At noon, canopy modified thermal comfort in the 5 m and 10 m ranges (r = −0.293 and r = −0.282, respectively, $p < 0.001$), but the effects grew weak in the 20 m range (Table 7). Green ground had limited cooling effects in the 5 m range, but the effects increased to significant strength (r = −0.415, $p < 0.001$ in 20 m radius) with increasing range. Pavement could strongly affect thermal comfort negatively (r = 0.487, $p < 0.001$ in 5 m radius), yet the effects decreased as the range increased while remaining significant. Water worked as a significant cooling material, but it only modified the thermal stress in the 5 m range (r = −0.335, $p < 0.001$). The relationship between roofs and PET was significant in the

5 m range (r = −0.314, $p < 0.001$). However, with increased range the relationship turned negative and a larger building area worsened thermal comfort. CR did not significantly affect thermal comfort, but when examining cooling effects in different directions the relationship was strong and significant. CR west had the most significant cooling effects (r = −0.399, $p < 0.001$ in 5 m radius), followed by CR south (r = −0.271, $p < 0.001$ in 5 m radius). CR on the north and east negatively affected thermal comfort in the 10 m and 20 m ranges, while the influence was weak.

**Table 7.** Pearson's correlation between ratio of garden elements and PET at noon.

| Garden Element | 5 m | 10 m | 20 m |
|---|---|---|---|
| Canopy | −0.293 *** | −0.282 *** | −0.193 *** |
| Green ground | −0.008 | −0.273 *** | −0.415 *** |
| Pavement | 0.487 *** | 0.447 *** | 0.345 *** |
| Water | −0.335 *** | −0.131 *** | −0.043 |
| Roof | −0.314 *** | 0.138 *** | 0.329 *** |
| CR | −0.322 *** | −0.176 *** | −0.05 |
| CR.N | −0.006 | 0.125 *** | 0.096 |
| CR.E | −0.125 *** | 0.003 | 0.153 *** |
| CR.S | −0.271 *** | −0.232 *** | −0.209 *** |
| CR.W | −0.399 *** | −0.393 *** | −0.276 *** |

$p < 0.001$ ***; green highlighted figures indicate significantly strong correlation (r ≥ |0.4|), light grey highlighted figures indicate significantly mild correlation (|0.2| ≤ r < |0.4|).

### 3.4.3. Relationship between Garden Configurations and Human Thermal Comfort in the Afternoon

In general, the relationship between PET and garden elements grew weaker in the afternoon compared to the relationship at noon (Table 8). Canopy modified thermal conditions of the resting areas, but only to a weak degree. Green ground cooling was only effective in the 20 m ranges (r = −0.248, $p < 0.001$). Pavement lowered thermal comfort in the 5 m and 10 m ranges (r = 0.262 and r = 0.256, respectively, $p < 0.001$). Water did modify the thermal comfort in the 5 m range, but the effects were weak, and the effects became nonsignificant with the range increases. The cooling effects of roofs were weak though significantly positive in the 5 m range and negative in 20 m, while the effect was not significant in the 10 m range. CR in the west had the most strong and significant effect (r = −0.237, $p < 0.001$ in 5 m radius), while CR in the other three directions did not significantly modify or only weakly affected thermal comfort.

**Table 8.** Pearson's correlation between ratio of garden elements and PET in the afternoon.

| Garden Element | 5 m | 10 m | 20 m |
|---|---|---|---|
| Canopy | −0.149 *** | −0.159 *** | −0.111 ** |
| Green ground | −0.008 | −0.153 *** | −0.248 *** |
| Pavement | 0.262 *** | 0.256 *** | 0.193 *** |
| Water | −0.174 *** | −0.07 | −0.011 |
| Roof | −0.15 *** | 0.082 | 0.184 *** |
| CR | −0.16 *** | −0.096 * | −0.033 |
| CR.N | 0.009 | 0.076 | 0.06 |
| CR.E | −0.015 | 0.014 | 0.047 |
| CR.S | −0.158 *** | −0.136 *** | −0.12 ** |
| CR.W | −0.237 *** | −0.227 *** | −0.146 *** |

$p < 0.05$ *, $p < 0.01$ **, $p < 0.001$ ***; light grey highlighted figures indicate significantly mild correlation (|0.2| ≤ r < |0.4|).

## 4. Discussion

### 4.1. Characteristics of Japanese Pond Gardens in Regard to Microclimate and Thermal Comfort

The study gardens create a thermally comfortable environment. Thermal comfort in the pond gardens was mostly "slightly warm", greatly ameliorated from thermal conditions

of urban open spaces assessed as "very hot" in this study. Although the microclimate in the gardens featured higher *RH* and limited air movement, the cooling effects of the gardens were manifested in decreased *Ta* and *Tg*. However, the difference in mean *Ta* between the study gardens and corresponding open areas appeared small (ranged from 0.03 °C to 2 °C) compared to previous studies. In Lisbon, Oliveira et al. [47] obtained as high as 6.9 °C *Ta* difference between urban parks and urban surroundings. Spronken-Smith and Oke [26] also reported that *Ta* difference between urban greens in Sacramento and surrounding areas could reach up to 7 °C if the green spaces are well irrigated. The studies both stressed the importance of evapotranspiration capacity in cooling, and claimed a high evapotranspiration rate could greatly contribute to green-space cooling. On the other hand, Chang et al. [29] found parks in Taipei were on average 0.81 °C cooler than their surroundings at noon in summer. The authors explained that the relatively little difference in *Ta* was due to low rate of evapotranspiration influenced by high humidity. For the same reasons, evapotranspiration in the study gardens was estimated to be weak, as *RH* in the gardens (66.9% on average) was very high. Limited air movement (0.24 m/s on average, nearly negligible) in the study gardens might be another factor that potentially suppressed evapotranspiration rate. Previous studies showed a low *v* could cause low vapor pressure deficit and accordingly decrease the evapotranspiration rate [48–50].

The results showed that various levels of thermal conditions existed in the gardens, making some areas more comfortable than others. Thermal comfort of resting areas such as S2 and C3 ranged from "comfortable" to "slightly warm", when other areas could become "warm" and "hot" occasionally. It is noteworthy that the less-comfortable resting areas all faced south and had low surrounding tree canopy. Although shaded by roofs, the areas received more longwave radiation emitted from the gardens, resulting in higher *Tg*, which in contrast could create favorable conditions during cold seasons. We assume that the resting areas facing south might be designed to adapt to cold seasons, since the highly shaded resting areas are likely to become uncomfortable during wintertime. With various space designs in one garden, people can choose comfortable areas to stay across seasons. Further study is needed to examine whether resting areas in traditional gardens were designed variously to meet different needs.

*4.2. Japanese Pond Garden Configurations and Their Cooling Effects*

A highly shaded environment was the most common characteristic of the pond gardens. Nearly three-quarters of areas around resting areas were covered by roofs or canopies. In line with previous studies [23,24], the results of this study emphasized the critical role shading plays for garden microclimate (Table 7). Besides the importance of shading, this study found that the degree of shading and its distribution were key to optimize its cooling effects. Results showed cooling effects of shade were restricted to the 5 m to 10 m ranges. Moreover, shades on the north or east could not effectively reduce thermal stress; only roofs and canopy on the west and south contributed to thermal comfort.

Regarding the shading materials, the resting areas were shaded mostly by roofs and occasionally by trees. This study did not specifically assess efficiency of shading materials for thermal comfort, but according to the correlation-analysis results, we presume cooling effects of the canopy might exceed effects of roofs. Correlation between roof area ratio and PET changed sharply from 5 m radius to 20 m radius, from positive effects to negative effects. The correlation between canopy area ratio and PET decreased gradually with the increase in distance, and the effects were consistently positive and significant. This might be because the heat stored by built-up areas at a large range surpasses the positive effects from shading. In contrast, trees provide various benefits in addition to shading effects such as reducing ambient air temperature with evapotranspiration.

Water can significantly affect thermal comfort, but its influence on people's thermal comfort appears to be restricted to a distance of about 5 m. Latent heat loss is seen as a distinctive cooling effect of water bodies, but high humidity in the study gardens could have impeded evaporation rate of ponds and limited the cooling-effect extension [51,52].

Low surface temperature of ponds probably contributed more to thermal comfort than evaporation effects. It is also likely that ponds adversely affect thermal comfort in calm environments by increasing humidity. This concern was reflected in the study gardens since, being pond gardens, the ratio of water bodies around the resting areas was generally small. Referring to the limited number of studies regarding Chinese gardens in Yangtze region, pavilions in Chinese gardens are often located close to ponds or even surrounded by ponds [36,53], whereas ponds in Japanese gardens were likely to be designed with some distance from people.

### 4.3. Implications for Planning and Design of Urban Green Spaces

Thermal conditions of urban environments in Kyoto during summer days, if left without microclimate modification, were demonstrated to be extremely uncomfortable, discouraging people from spontaneous activities [22]. To promote green-space uses in hot and humid regions, effective climate-responsive design is necessary. Based on our findings regarding how traditional gardens were designed to alleviate people's thermal stress, we suggest the following implications for urban planners and designers.

First, providing enough shade matters. Radiation is the largest threat to people's thermal comfort, but it can be well controlled by landscape design. The design of study gardens indicates that shade should be created with a focus around the areas people use. In humid regions, wind also significantly improves thermal comfort by accelerating evaporation on people's skin. However, in contrast to shade, which can be easily and precisely created at desired areas, air movement is generated in a complex mechanism and is usually determined by urban morphology and local climate. Therefore, creating shade is the most efficient strategy in urban design.

Second, shade must be created with precision, as plenty of shade may not be enough. Shading can benefit people only when the shade is casted precisely on areas people frequently use, such as benches and walkways, because shading effects are restricted to short distances. Park designs featuring numerous trees but unshaded benches unfortunately are commonplace. As Eliasson [54] argued, climate-responsive design is frequently underestimated by landscape designers. Functionality and esthetic perspectives are often prioritized over thermal comfort, risking underused green spaces. We suggest urban designers and planners allocate resting areas close to shading elements, keeping in mind that the west and south sides are the most-effective directions to place shading elements. Moreover, people in resting areas are more sensitive to thermal conditions than people in playgrounds or walking on walkways. Not only can people who are walking and exercising reduce some heat energy by sweating and therefore become more tolerant to heat, people exercising and playing sports are usually strongly motivated to engage in these activities, and therefore feel less dissatisfied with thermal conditions [55]. In contrast, resting areas with uncomfortable conditions are more likely to make people feel dissatisfied as the rest sought eludes them.

Third, ground surface should be designed carefully and flexibly. Water is an attractive element in landscaping. Apart from its aesthetic value, the cooling effect of water is highly valuable. However, this study indicates that water bodies improve human thermal comfort only in a small range, within a 5 m distance. Moreover, water bodies could adversely affect thermal comfort by increasing humidity, particularly in humid regions. To take advantage of the cooling effects of water, water bodies should be close to areas where people sit or stay, and in areas with frequent air movement or low humidity. Among the garden elements we tested, pavement has the strongest negative impact on thermal comfort. We suggest using more green surface materials such as lawns or small shrubs, or permeable ground instead. When designing urban parks or open areas, large areas of hardened surface for activities may be required. In that case, creating sufficient shade over the pavement might significantly lower the negative effects by lowering the surface temperature of pavement [56]. The use of advanced paving, such as permeable paving

and paving materials with a high reflectivity and emissivity, is also an efficient strategy for heat-stress mitigation [57].

Last but not least, integrated policy is required to improve green-space thermal conditions. Urban parks in Japan are required to be distributed in neighborhoods at a maximum 250 m walking distance. However, there is no requirement for a minimum green-coverage ratio. To our knowledge, no studies have comprehensively investigated thermal conditions of urban parks in Japan. But judging from the authors' observation, many urban parks in Kyoto are undershaded, and therefore cannot provide safe green spaces for residents during the hot seasons. Such conditions prohibit residents from enjoying activities outdoors or interacting with nature. We believe climate change necessitates that policymakers improve regulations for green-space design in order to guarantee urban green spaces are designed to be safe and beneficial to people's well-being. Further study examining thermal conditions of urban parks might be helpful for policymakers to recognize the problems in urban parks and the urgent need for improving the quality of urban green spaces.

### 4.4. Limitations

Due to the exploratory nature of this study and a number of restrictions placed on the investigation because of the nature of the study sites, several limitations need to be considered when interpreting the results. First, the thermometer used was not radiation-shielded, potentially resulting in a higher air temperature and PET values, particularly for the parts of the open areas fully exposed to solar radiation. However, while meteorological parameters in the open areas were collected as a reference, the data were not used in, and thus did not affect, the analysis of the gardens' cooling effects or the larger study conclusions. Regarding the study points in the gardens, owing to the areas being mostly shaded by roofs and trees, the meteorological parameters recorded in the gardens were unlikely to be exacerbated to a degree that would preclude gaining a general understanding of microclimate conditions of the gardens. Due to irregular public events in the gardens and the limited number of instruments, simultaneous surveys in the three gardens were prevented. To minimize bias when comparing microclimate conditions among the three gardens, the survey was conducted under similar weather conditions to the largest extent possible.

Simplified configuration analysis is another limitation in this study. Species of trees, pavement materials, surface materials that are in the sun or in the shade, and orientation of resting areas would all presumably affect thermal comfort in resting areas, but were not the target of this exploratory study. As a result, the correlation between thermal comfort and garden configurations appeared weak. As is frequently the case with empirical research, not all factors could be controlled during the data collection or included in the analysis. However, it was possible to identify potential factors that significantly affected people's thermal comfort. Further research will be vital to better understand the interaction between garden design and its effects on human thermal comfort.

The number of samples (six investigation points in three gardens) did not allow generalizing characteristics of traditional pond gardens in terms of thermal conditions and designs. Following up on this exploratory study with further investigations in a larger number of traditional gardens existing in Kyoto would be valuable to gain a more comprehensive understanding of climate-responsive strategies of traditional gardens. In particular, investigations in different seasons would be helpful to clarify how the gardens work across extreme variations in temperature. We also would like to stress that the microclimate of the study gardens is influenced by their surrounding environment. Effects from the nearby forests are certainly relevant. Even though the thermal conditions of open areas in this study showed the study area to be extremely uncomfortable, general climatic conditions of the area could be cooler and more humid than that in the city center. Again, further research is required to explore effective design strategies in extremely urbanized areas.

## 5. Conclusions

To our knowledge, this is the first study to investigate microclimate and human thermal comfort in Japanese gardens. We aimed to provide useful guidance on climate-responsive green-space design by studying strategies from traditional gardens. Investigations of microclimates carried out on hot summer days in Japanese pond gardens demonstrated that the pond gardens modified "very hot" local thermal conditions to "slightly warm" conditions. Among the areas investigated in the gardens, some areas were more comfortable than others, with the difference in thermal comfort likely related to the difference in spatial configurations.

Garden elements in the gardens were demonstrated to influence thermal comfort in various ways on different occasions. Firstly, a high ratio of shade (around 75%) appeared most critical in creating comfortable microclimates on hot and humid summer days. In terms of shading materials, both tree canopy and roofs were able to improve thermal conditions, but the cooling effects of tree canopy extended further, while that of roofs was limited to a 5 m radius. Distribution of shading was also important. Shading on the north and east had limited influence on thermal comfort, and occasionally negative influence on human thermal comfort. Therefore, it is important to place trees or artificial shades on the south and west of green spaces or other areas that people use.

Secondly, ground-surface materials (water, green ground, and pavement) affected thermal comfort in very different manners. Water bodies improved thermal comfort within a small range, while green ground improved thermal comfort across a large range. Pavements affected thermal comfort negatively, with effects extending up to 20 m. It remained unclear why cooling effects of water bodies and green ground was restricted to a certain range, but high humidity of the study sites potentially influenced cooling intensity of surface-cover materials. Thermal performance of surface materials appeared diverse depending on the surrounding environment and climatic context. Urban designers should thus be clearly aware of site context prior to designing sites. Overall, a better understanding and evidence on cooling mechanisms of garden elements under different climatic contexts is necessary to improve urban designers' and planners' decisions and designs.

Lastly, this study contributes to the literature by demonstrating how many valuable climate-responsive strategies can be inspired by traditional gardens. We believe space design of traditional gardens was developed deliberately to modify local climate and provide a comfortable environment for its users. Nevertheless, only a very small number of studies have explored the microclimate of traditional gardens, the large number of traditional gardens in the world notwithstanding. Broadening and deepening our understanding of microclimate and thermal comfort of traditional gardens across various countries and regions thus remains a vital task in a world growing perpetually hotter.

**Supplementary Materials:** The following are available online at https://www.mdpi.com/2071-1050/13/5/2736/s1, File S1: Garden elements area, File S2: Meteorological data.

**Author Contributions:** Conceptualization, L.C. and S.S.; methodology, L.C. and S.S.; software, L.C.; validation, L.C.; formal analysis, L.C.; data curation, L.C.; writing—original draft preparation, L.C.; writing—review and editing, L.C., C.D.D.R., and S.S.; visualization, L.C.; supervision, S.S.; funding acquisition, S.S. All authors have read and agreed to the published version of the manuscript.

**Funding:** This study was supported by JSPS KAKENHI grant number 18H02226. Parts of this research were supported by the FEAST Project (C.D.D.R., no. 14200116), Research Institute for Humanity and Nature (RIHN).

**Institutional Review Board Statement:** Not applicable.

**Informed Consent Statement:** Not applicable.

**Data Availability Statement:** The data presented in this study is contained within the supplementary materials.

**Acknowledgments:** The authors deeply appreciate three Japanese gardens Shoren-in, Chion-in and Joju-in, for providing valuable opportunity to conduct the survey in the beautiful gardens. We would like to thank Kojima and Zhang for sharing knowledge on Japanese gardens. The authors also want to thank Wang, Morita, Oka, Shinmura, Yagura, Sugano, Shoda, Sun, and Tan for their kind assistance with the field measurements.

**Conflicts of Interest:** The authors declare no conflict of interest.

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
