# Peer review of "Climate-Responsive Green-Space Design Inspired by Traditional Gardens: Microclimate and Human Thermal Comfort of Japanese Gardens"

_sustainability, doi:10.3390/su13052736_

Round 1

Reviewer 1 Report

The presented research contributes to the knowledge on climate responsive strategies for the design of public green spaces, in specific conditions of hot and humid climate, learning from tradition and long-standing knowledge of Japanese garden art.

The research path is clearly explained, including research limitations. I agree with the authors, that further research would be vital to understand the interaction between garden design and its effects on human thermal comfort, including investigations in different seasons.

I suggest expanding  the literature review and adding references to studies of thermal comfort in traditional gardens, for example:

Sihan Xue, Yiqiang Xiao, Study on the Outdoor Thermal Comfort Threshold of Lingnan Garden in Summer, Procedia Engineering, Volume 169, 2016, Pages 422-430, ISSN 1877-7058, https://doi.org/10.1016/j.proeng.2016.10.052.

Ojaghlou, M., Khakzand, M. (2019). Thermal Comfort Characteristic of 5 Patterns of a Persian Garden in a Hot-Arid Climate of Shiraz, Iran, Journal of Landscape Ecology, 12(3), 1-33. doi: https://doi.org/10.2478/jlecol-2019-0016

Author Response

We thank the reviewer for constructive and helpful advice on this manuscript. 

Comments Response Line numbers
I suggest expanding the literature review and adding references to studies of thermal comfort in traditional gardens.

We thank the reviewer for commenting on the contributions of this manuscript. 

As the reviewer kindly pointed out, this study comes with many limitations and we believe further improvement is required to obtain a comprehensive understanding on Japanese garden microclimate. Nevertheless, this manuscript indicates local-based valuable climate responsive strategies could be learned from traditional gardens. We hope more investigations in traditional gardens will be carried out in the future. 

I suggest expanding the literature review and adding references to studies of thermal comfort in traditional gardens.

We highly appreciate the reviewer’s recommendation on references. Both of the articles have been carefully reviewed and we have expanded the literature review as the reviewer suggested.

89-94

Reviewer 2 Report

CONGRATULATIONS ON A CAREFULLY DESIGNED AND CARRIED OUT STUDY, WITH HIGH QUALITY REPORTING. I suggest proof-reading to deal with some syntax issues.

Author Response

Comments

Response

CONGRATULATIONS ON A CAREFULLY DESIGNED AND CARRIED OUT STUDY, WITH HIGH QUALITY REPORTING.

We thank the reviewer for the positive response to this manuscript.

I suggest proof-reading to deal with some syntax issues.

Thank you for the suggestion. In response to the reviewer's comments, we have proofread the manuscript again.

Reviewer 3 Report

Questions to clarify: 1. The gardens/case studies are located at the foot of mountains in the neighborhood of some large green (forested?) areas, less in the extremly dense built urban fabric. Even the authors wrote „Most of the solar radiation in the morning is thus blocked by the hill” (Chapter 3.4.1., page 14). Do you think this location doesn’t affect considerable the authenticity of the results? 2. The microclimate and thermal comfort of green areas depends widely from the size of the gardens/parks. The size of the gardens in study would be anyway an important beckground information. 3. The microclimate and thermal comfort of green areas is influenced by other factors as well, which are not studied and presented for the whole gardens (in general) just for the „study points” and their proximity (analyzed only in the 20 m radius of study points). In this context the canopy coverage procent of the whole gardens, some characteristics of vegetation, for ex. the typical vegetal layers (lawn, shrubs, tree canopy) are also needed to have an adequate overview of the general park conditions In general I consider it a good and original article with a strong bibliographical bakground and very current resarch questions. I support the publication after a few minor complementations mentioned above.

Author Response

Comments

Response

Line numbers

The gardens/case studies are located at the foot of mountains in the neighborhood of some large green (forested?) areas, less in the extremely dense built urban fabric. Even the authors wrote „Most of the solar radiation in the morning is thus blocked by the hill” (Chapter 3.4.1., page 14). Do you think this location doesn’t affect authenticity of the results?

We thank the reviewer for raising this point. The study gardens are located in an area that is highly urbanized and also is very close to forests. Such urban conditions are common in Kyoto, a small city surrounded by mountains on three sides and inhabited by a large population.

We agree that the microclimate in the gardens is influenced by nearby forests, and thermal comfort in this area is better than in the city center. This is why we measured meteorological data in open areas near the gardens. The manuscript indicates that, even though the study area is close to forests, regular urban open spaces in summer are still uncomfortable but gardens can effectively modify their microclimate and mitigate people’s thermal stress. However, green spaces in the city center might require a different selection of design strategies as climatic conditions are likely to be hotter and drier in extremely urbanized areas with less greenery. We added corresponding background explanations in the discussion section. 

597-603

2. The microclimate and thermal comfort of green areas depends widely from the size of the gardens/parks. The size of the gardens in study would be anyway an important background information.

We thank the reviewer for the advice. Sizes of the study gardens are added in the manuscript, see Table1.

169

3. The microclimate and thermal comfort of green areas is influenced by other factors as well, which are not studied and presented for the whole gardens (in general) just for the „study points” and their proximity (analyzed only in the 20 m radius of study points). In this context the canopy coverage percent of the whole gardens, some characteristics of vegetation, for ex. the typical vegetal layers (lawn, shrubs, tree canopy) are also needed to have an adequate overview of the general park conditions. 

We concur with the reviewer that information about the landscaping characteristics of the gardens is important. Thank you for the constructive suggestion. We have added the canopy coverage ratio, pond area ratio, dominant tree species, and ground surface materials to Table 1. 

169

In general I consider it a good and original article with a strong bibliographical background and very current research questions. I support the publication after a few minor complementations mentioned above.

We appreciate the reviewer’s positive evaluation of our study.